# The Impact of Sex on the Neurocognitive Functions of Patients with Parkinson’s Disease

**DOI:** 10.3390/brainsci11101331

**Published:** 2021-10-09

**Authors:** Mei-Ling Chen, Chun-Hsiang Tan, Hui-Chen Su, Pi-Shan Sung, Chia-Yi Chien, Rwei-Ling Yu

**Affiliations:** 1Institute of Behavioral Medicine, College of Medicine, National Cheng Kung University, Tainan 701, Taiwan; mouchue192@gmail.com (M.-L.C.); a0918732736@gmail.com (C.-Y.C.); 2Department of Neurology, Kaohsiung Medical University Hospital, Kaohsiung Medical University, Kaohsiung 807, Taiwan; chtan@kmu.edu.tw; 3Graduate Institute of Clinical Medicine, College of Medicine, Kaohsiung Medical University, Kaohsiung 807, Taiwan; 4Department of Neurology, National Cheng Kung University Hospital, College of Medicine, National Cheng Kung University, Tainan 701, Taiwan; n510807@mail.hosp.ncku.edu.tw (H.-C.S.); em75482@email.ncku.edu.tw (P.-S.S.)

**Keywords:** Parkinson’s disease, sex, neurocognitive impairment

## Abstract

This study aimed to understand the impact of sex on the neurocognitive function of patients with Parkinson’s disease (PD). Ninety-four participants with idiopathic PD and 167 age-matched healthy individuals as normal controls (NCs) were recruited and underwent comprehensive neuropsychological assessments. Sex differences were found in NCs, but not in patients with PD. Among male participants, patients with PD showed worse performance on the Digit Symbol Substitution (DSS) (*p* < 0.001) test and Symbol Search (SS) (*p* < 0.001) than NCs. Among female participants, patients with PD showed worse performance on the category score of the Modified Wisconsin Card Sorting Test (*p* < 0.001), SS (*p* < 0.001), and pentagon copying (*p* < 0.001) than NCs. After controlling for the effects of age and years of education, Hoehn and Yahr stage was found to predict the performance of the Color Trails Test part A (βA = 0.241, pA = 0.036), Stroop Color and Word Test (β = −0.245, *p* = 0.036), and DSS (β = −0.258, *p* = 0.035) in men with PD. These results indicate the differential effect of sex on the neurocognitive function among healthy aging and PD populations. The disappearance of sex differences, which is present in healthy aging, in patients with PD suggests a gradual loss of the neuroprotective effect of estrogen after the initiation of the neurodegenerative process. This study also found mental flexibility and visuospatial function to be the susceptible cognitive domains in women with PD, while the disease severity could predict the working memory and processing speed in men with PD.

## 1. Introduction

Parkinson’s disease (PD) is the second most common neurodegenerative disease worldwide. The prevalence of PD increases with age, with greater prevalence in men [1,2]. The pathological hallmark of PD is the aggregation of Lewy bodies and degeneration of dopaminergic neurons. The involvement of the nigrostriatal pathways reduces striatal dopamine levels and causes various motor and non-motor symptoms (NMS) [3]. Among the NMS, neurocognitive dysfunction causes a decline in the quality of life of patients with PD. Full-blown dementia, as the final stage of neurocognitive impairment, leads to the impairment of self-care functions, a burden on caregivers, and surging health care costs [4,5]. Previous studies have shown that neurocognitive dysfunction in PD can develop across various domains (e.g., executive function, attention, processing speed, visuospatial ability, memory, and language) [4,6,7,8].

In the process of normal aging, the influence of sex in causing differences in neurocognitive function has been reported, with women showing better performance in verbal-related tests [9] and men in visual-spatial tests [10]. However, some studies found no sex difference in the general neurocognitive function between male and female PD [11,12,13]. Nevertheless, other studies reported that male patients have more subjective complaints [14], lower global cognitive function in the early stage [15,16,17], and more mild cognitive impairment [4,18,19]. In addition, the progress to mild cognitive impairment was reported to be more rapid in men with PD than in women [4,19]. In contrast, some studies suggested that women with PD patients have poor general cognitive function [20].

Furthermore, previous investigations have suggested that the domains involved in the neurocognitive dysfunction of PD may be sex-specific, although discrepancies between different studies exist, which can result from the differences in the assessment tools applied or variations in the disease stage of the patients recruited. In addition, the lack of healthy normal controls (NCs) in previous studies precludes a clear elucidation of whether the impact of sex on the neurocognitive functions of PD patients originate from sex or disease. Moreover, clinical characteristics (e.g., onset age, disease duration, disease severity, and LED) are associated with neurocognitive function [4,18]. Taking clinical characteristics into consideration may help elucidate the heterogeneity of PD besides sex.

Thus, to elucidate the impact of sex on the neurocognitive function of patients with PD, we cross-sectionally evaluated the neurocognitive function of both healthy aging participants and patients with PD across different disease stages, including those in the later stages. In addition, we further explored the relationship between neurocognitive function and clinical characteristics, including the age of onset, disease duration, levodopa equivalent dose (LED), and Hoehn and Yahr Staging Scale (H&Y stage) of patients of each sex.

## 2. Materials and Methods

### 2.1. Participants

A total of 94 patients with PD and 167 healthy participants as NCs were recruited. All patients were diagnosed with PD by experienced board-certified neurologists according to the United Kingdom PD Society Brain Bank clinical diagnostic criteria, with motor symptom onset after 50 years of age. The NCs were recruited from communities. All participants’ exclusion criteria were as follows: atypical features of parkinsonism, history of brain operations, severe systemic diseases, psychiatric diseases (e.g., depression and schizophrenia), or illiteracy. Atypical features of parkinsonism denote symptoms or signs suggestive of disorders or syndromes other than PD, including evident cerebellar abnormalities, downward vertical supranuclear gaze palsy, selective slowing of downward vertical saccades, diagnosis of probable behavioral variant frontotemporal dementia or primary progressive aphasia within the first five years of disease, parkinsonian features restricted to the lower limbs, clinical course consistent with drug-induced parkinsonism, poor response to levodopa, cortical sensory loss, ideomotor limb apraxia, normal functional neuroimaging of the presynaptic dopaminergic system, rapid progression of gait disturbance requiring regular use of a wheelchair or severe autonomic failure within the first five years of onset, early bulbar dysfunction, recurrent falls within the first three years of onset, disproportionate dystonia or contractures of hand or feet within the first ten years, and unexplained pyramidal tract signs. These atypical features follow the diagnostic criteria proposed by the movement disorder society [21].

Informed consent was obtained from all participants following the ethical standards outlined in the 1964 Declaration of Helsinki, and the Institutional Review Boards of the National Cheng Kung University Hospital (approval number: A-ER-107-425) and Kaohsiung Medical University Hospital (approval number: KMUHIRB-G(II)20160001) approved the study protocols.

### 2.2. Assessment

#### 2.2.1. Demographic and Clinical Characteristics

We collected data on the age and years of education of patients with PD and NCs, age of onset, disease duration, Hoehn and Yahr Staging Scale, and LED of patients with PD.

#### 2.2.2. Neuropsychological Assessment

We evaluated six neurocognitive domains: executive function, memory, processing speed, visuospatial ability, attention, and language. Detailed neuropsychological assessment tools used for the evaluation of each cognitive domain are presented in Table 1.

### 2.3. Statistical Analysis

All statistical analyses were conducted using IBM SPSS Statistics 22 (SAS Institute Inc., Cary, NC, USA). All variables were tested for normal distribution using the Kolmogorov–Smirnov test. The study groups were compared using the t-test and Mann–Whitney U test for continuous variables and the chi-square test or Fisher–Freeman–Halton exact test for categorical variables. Significance tests were two-tailed, with *p* < 0.05. A Bonferroni correction for multiple comparisons was applied to decrease the likelihood of a type II error, resulting in the adoption of 0.002 (i.e., 0.05/25) as the cutoff for statistical significance.

## 3. Results

To understand the neurocognitive deficits attributable to the neurodegenerative course of PD, we compared the neurocognitive performance between patients with PD and NCs. As shown in Table 2, we found that patients with PD had worse performance on the category score of the Modified Wisconsin Card Sorting Test (M-WCST, *p* < 0.001), total score of Category Fluency (*p* < 0.001), word score of Stroop Color and Word Test (SCWT) (*p* < 0.001), scaled score of Digit Symbol Substitution (DSS, *p* < 0.001), scaled score of Symbol Search (SS, *p* < 0.001), pentagon copying(*p* < 0.001), scaled score of Block Design (BD, *p* = 0.002), and raw score of Logical Memory (LM, *p* = 0.002) than NCs.

In addition, to understand the difference in the neurocognitive functions attributable to the effect of sex among patients with PD and NCs, we compared the neurocognitive function between men and women in the two study groups. Furthermore, the participants’ sex-stratified analysis was used to further assess the neurocognitive function between men and women among the PD and NC groups. The comparisons between sexes in each study group and the participants’ sex-stratified analyses are shown in Table 3. In the NCs group, men had received significantly longer years of education than women; thus, we controlled for the effect of years of education during our analysis. After adjusting for years of education, female NCs were found to have a higher total score of Category Fluency (*p* < 0.001), the word and color score of SCWT (*p* < 0.001), the raw score of LM immediate (*p* = 0.001) and delayed recall (*p* = 0.002) than their male counterparts. However, no difference in neurocognitive functions was found between men and women with PD. Among male participants, patients with PD performed worse on the Color Trails Test (CTT) part A (*p* < 0.001) and DSS (*p* < 0.001) than the NCs, indicating a worse processing speed in male patients with PD than male NCs. Furthermore, among female participants, patients with PD performed worse on the category score of M-WCST (*p* < 0.001), DSS (*p* < 0.001), and pentagon copying (*p* < 0.001) than the NCs, indicating worse executive function, processing speed, and visuospatial ability in female patients with PD than in female NCs.

Moreover, we performed a further regression analysis of the relationship between clinical characteristics (e.g., age of onset, disease duration, LED, and H&Y stage) and neurocognitive function in male and female patients with PD, respectively.

The correlations between the clinical characteristics and neurocognitive function are shown in Table 4 and Table 5. The regression results showed that (Table 6), with the age and years of education controlled, H&Y stage could predict the performance of working memory (CTT part B, βB = 0.222, pB = 0.046), and processing speed (CTT part A, βA = 0.241, pA = 0.036; the color score of the SCWT, β = −0.245, *p* = 0.036; DSS, β = −0.258, *p* = 0.035) in male patients with PD. However, no significant associations between the clinical characteristics and neurocognitive function were found in female patients with PD.

## 4. Discussion

In the present study, we investigated the effect of sex on differences in the neurocognitive function of patients with PD. We found that patients with PD had significantly worse performance in executive function, processing speed, visuospatial function, and memory function than their normal control counterparts, while no significant differences were found in the language and attention functions between the patients with PD and the NCs. These findings are partially compatible with those of the previous studies, which showed that patients with PD started to develop neurocognitive deficits in the early stage [4,5,7,8], while language function is relatively preserved [6,7,8]. However, in the present study, we did not find a significant difference in the attention function of patients and healthy aging individuals, which seems to be inconsistent with the results of previous studies [7,8,22].

On the other hand, this discrepancy may arise from the nature of attention and the tasks applied to assess attention. In the present study, we used tools (i.e., information registration, serial seven, and digit span) designed to measure simple rather than complex attention functions. Previous studies suggested that patients with PD have impaired complex attention but relatively preserved, simple attention [23,24]. Previous studies have suggested that the major determinants of neurocognitive dysfunction depend on the capacity of the supervisory attentional system [25]. Suppose the task demands are within the capacity of the attentional resources of patients with PD. In that case, patients may have similar performance in terms of attention as that of healthy aging individuals. However, once the task’s attention demands overwhelm the patient’s attentional resources, additional internal cues and mental operation will be required, in which patients with PD may not perform as well as their age-matched counterparts [26].

In this study, we studied the neurocognitive function of both sexes in healthy aging and PD groups to understand the impact of sex on neurocognition. Our findings are compatible with previous results in healthy aging populations [9,10,27]. We found that healthy aging women’s verbal fluency, processing speed, and verbal memory were better than the aging performance of healthy men. In addition, there was no significant difference in executive function, attention function, and language between men and women in the healthy aging populations.

Some studies have suggested the potential impact of sex on the neurocognitive function of patients with PD. Female patients with PD were found to perform better in terms of verbal fluency [13,16,17,28], verbal memory [12,29], and processing speed than male patients with PD [13,15,16], while male patients with PD performed better on visuospatial ability tests [16,28,29,30]. However, in our study, male and female patients with PD showed no difference in any of the neurocognitive domains examined in the current study. These diverging results may be due to the fact that most patients with PD recruited in the previous studies had shorter disease duration [12,13,16,17,28,29] or intact neurocognitive function [13], while we recruited patients in the later stages of the disease as well. Moreover, the lack of sex differences in the neurocognitive function of patients with PD and the presence of sex differences in the healthy aging population indicates the differential effect of sex between healthy aging and the neurodegeneration of PD, which may be because of the protective effect of estrogen, which has been shown to be neuroprotective and reduces the risk of PD in women [31]. Estrogen protection may still exist in the early disease stage but is gradually lost in the later stage [32]. The differential effect of estrogen is supported by the evidence showing its neuroprotective effect on striatal degeneration or maintenance of neurocognitive integrity [33,34,35] but not in the damaged striatum [36]. In other words, when neurodegeneration gradually progresses into later stages, the protective effect of estrogen may become weaker or lost. The other possible mechanism for the differential effect of sex may arise from the innate differences in dopamine concentration between men and women. Women with PD have higher dopamine concentrations than men with PD, which explains the generally more benign clinical characteristics found in women than men in the early disease stage [1,2,37]; however, once dopamine depletion in the striatum exceeds the innate difference conferred by sex, the difference in the neurocognitive dysfunction of male and female patients with PD is abolished [32].

We further applied sex-stratified analysis to explore the impact of the disease on neurocognitive functions, specifically in male and female patients with PD. Among female participants, we found that the visuospatial ability of patients with PD is worse than that of NCs, which is consistent with the results of a previous study [38]. However, we further found that female patients with PD performed worse in terms of executive function (i.e., cognitive flexibility) than the female NCs, while the previous study did not find any such difference. The difference in the results may be due to the younger and the earlier disease stage of the PD patients recruited in the previous study [38]. The results of this study may be more representative of the patients with PD from all stages of the neurodegenerative course. Our findings suggest that the cognitive flexibility of executive function and visuospatial ability are the susceptible cognitive domains in female patients with PD and have a higher risk of being damaged, which may result from the loss of estrogen protection.

Moreover, growing evidence has also shown the development of a host of socioaffective troubles in addition to neurocognitive dysfunctions in the disease course of PD [39]. Our previous study also found susceptibility to social cognition in female patients with PD [40]. It is possible that the mechanism underlying the more severe decline of the socioaffective functions is similar to the effect of sex on the neurocognitive dysfunctions observed in this study. As the socioaffective troubles of patients with PD also cause impairment in the quality of life and have to be taken into consideration for a complete understanding of the impact of the disease on patients.

Previous studies have shown that the clinical characteristics (e.g., onset age, disease duration, LED, and H&Y stage) of patients with PD are associated with neurocognitive function [4,18,28]. In the present study, we found a differential effect of sex on specific neurocognitive functions. Moreover, disease severity (H&Y stage) was found to predict mental shifting ability and processing speed in male patients with PD; however, no clinical features were found to be predictive factors in female patients with PD. Although female patients with PD are a minority group, there is a lack of research on these groups. Future studies are needed to investigate the modulatory factors of the neurocognitive functions of female patients with PD.

The current study has some limitations. First, the lack of accurate biological markers representing the estrogen level limits the extrapolation of the neuroprotective effect of estrogen on cognition in the female healthy aging group. Moreover, the mechanism of the protective effect of estrogen on cognitive function remains unclear, and further studies are needed. Second, the diverse heterogeneity of motor symptoms makes it difficult to accurately analyze the association with the neurocognitive performance of patients with PD. However, this topic is beyond the scope of this study. Therefore, we suggest future studies to include motor symptom subtypes for a more accurate prediction of neurocognitive function performance in patients with PD.

## 5. Conclusions

The present study found a differential effect of sex on the neurocognitive function of healthy aging and PD population. The poor performance of both male and female patients with PD in neurocognitive function tests and the disappearance of the sex difference, which is present in the healthy aging population, in patients with PD suggest a gradual loss of the neuroprotective effect of estrogen after the initiation of the neurodegenerative process. Furthermore, processing speed was the cognitive function domain most susceptible to neurodegeneration, regardless of sex, while mental flexibility and visuospatial deterioration were the vulnerable functions in female patients with PD. The results of this study may provide a deeper understanding of the impact of sex on the neurocognitive function of patients with PD and may help elucidate the effect of sex on the process of neurodegeneration.

## Figures and Tables

**Table 1 brainsci-11-01331-t001:** The neuropsychological tests used in the current study.

Domain	Neuropsychological Tests
Executive Function	Modified Wisconsin Card Sorting Test (M-WCST): the number of categories achieved, and perseverative errors Stroop Color and Word Test (SCWT): color-word score Category Fluency (fruit, fish, and vegetable) Color Trails Test (CTT)-part B Similarities ^a^ Matrix reasoning ^a^
Attention	Attention Test ^†^ Digit Span ^a^
Processing Speed	SCWT: word score and color score CTT-part A Digit Symbol Substitution ^a^ Symbol Searching ^a^
Visuospatial ability	Pentagon copy ^†^ Block design ^a^
Memory	Logical Memory (LM) ^b^ Visual Reproduction (VR) ^b^
Language ^†^	Naming Repetition Verbal comprehension

^a^ Subtest of Wechsler Adult Intelligence Scale-Third Edition (WAIS-III); ^b^ Subtest of Wechsler Memory Scale-Third Edition (WMS-III); ^†^ Subtest of Mini-Mental State Examination (MMSE).

**Table 2 brainsci-11-01331-t002:** The demographic data and neurocognitive function in study groups.

	PD (*n* = 94) Mean (SD)	NC (*n* = 167) Mean (SD)	*p* Value
age, y	63.96 (6.17)	64.88 (8.54)	0.320 ^c^
Education, y	12.10 (4.05)	11.94 (3.60)	0.393 ^c^
**Executive function**			
M-WCST-C	4.38 (2.04)	5.56 (1.65)	<0.001 ^c,^*
M-WCST-P	5.77 (8.49)	3.42 (4.80)	0.024 ^c^
SCWT-color word score	29.12 (11.69)	32.50 (10.99)	0.041 ^c^
Category Fluency	34.04 (8.26)	38.73 (8.83)	<0.001 ^c,^*
CTT-B	136.15 (90.66)	108.21 (37.71)	0.013 ^c^
Similarities	10.66 (2.89)	11.78 (2.69)	0.002 ^c^
Matrix Reasoning	10.86 (2.96)	11.85 (2.89)	0.009 ^c^
**Attention**			
Attention test ^†^	7.28 (1.15)	7.61 (0.73)	0.028 ^c^
Digit Span	11.45 (2.62)	12.01 (2.77)	0.181 ^c^
**Processing speed**			
SCWT-word score	75.13 (20.07)	84.31 (17.89)	<0.001 ^b,^*
SCWT-color score	57.66 (265.969)	62.86 (14.02)	0.007 ^b^
CTT-A	69.14 (38.74)	52.23 (19.06)	<0.001 ^c,^*
Digit Symbol Substitution	10.16 (2.35)	12.10 (2.38)	<0.001 ^c,^*
Symbol Searching	10.35 (2.65)	12.31 (2.32)	<0.001 ^c,^*
**Visuospatial ability**			
Pentagon copy ^†^	12/82	3/164	<0.001 ^a,^*
Block Design	9.93 (2.67)	10.94 (2.75)	0.002 ^c,^*
**Memory**			
LM-I (r.s.)	30.03 (13.49)	35.46 (11.51)	0.002 ^c,^*
LM-II (r.s.)	17.43 (10.65)	21.84 (9.25)	0.001 ^b,^*
LM-recognition (r.s.)	22.64 (4.35)	24.49 (3.75)	0.001 ^c,^*
VR-I (r.s.)	67.05 (19.99)	74.67 (14.22)	0.005 ^c^
VR-II (r.s.)	44.99 (23.44)	52.36 (21.45)	0.019 ^c^
VR-recognition (r.s.)	41.16 (4.13)	42.24 (3.90)	0.027 ^c^
**Language** ^†^			
Naming	0/94	0/167	-
Repetition	1/93	4/163	0.657 ^a^
Verbal comprehension	2/92	3/164	>0.999 ^a^

^a^ Fisher Exact Test; ^b^
*t*-test; ^c^ Mann–Whitney U test; * *p* < 0.002 (Bonferroni correction for multiple comparisons); ^†^ Subtest of Mini-Mental State Examination (MMSE)Abbreviations: please see Table 1 and SD, standard deviation; r.s., raw score; LM-I, immediate memory of the logical memory; LM-II, delayed memory of the logical memory; VR-I, immediate memory of the visual reproduction; VR-II, delayed memory of the logical memory; M-WCST-C and M-WCST-P indicate the achieved categories and perseverative, respectively.

**Table 3 brainsci-11-01331-t003:** Sex-stratified analysis of the neurocognitive tests in PD and NCs.

	Male NCs (*n* = 53)	Female NCs (*n* = 114)	Male PD (*n* = 60)	Female PD (*n* = 34)	*p* ^w^ Value	*p* ^x^ Value	*p* ^y^ Value	*p* ^z^ Value
	Mean (SD)	Mean (SD)	Mean (SD)	Mean (SD)				
age, y	66.59 (8.92)	64.09 (8.28)	64.45 (5.62)	63.09 (7.04)	0.077 ^c^	0.307 ^b^	0.098 ^c^	0.760 ^c^
education, y	12.91 (3.31)	11.49 (3.65)	12.38 (4.06)	11.59 (4.05)	0.012 ^c,^*	0.254 ^c^	0.920 ^c^	0.514 ^c^
onset age, y	-	-	60.15 (5.86)	59.09 (6.78)	-	0.429 ^b^	-	-
Disease duration, y	-	-	4.47 (2.76)	4.19 (3.79)	-	0.249 ^c^	-	-
LED	-	-	557.53 (265.38)	463.92 (277.75)	-	0.085 ^c^	-	-
H&Y stage	-	-	1.50 (0.893)	1.26 (1.05)	-	0.329 ^b^	-	-
**Executive function**								
M-WCST-C	5.47 (1.69)	5.60 (1.63)	4.47 (2.054)	4.24 (2.05)	0.310 ^e^	0.585 ^c^	0.009 ^c^	<0.001 ^c,^*
M-WCST-P	3.81 (6.09)	3.24 (4.08)	5.43 (8.073)	6.35 (9.29)	0.226 ^e^	0.539 ^c^	0.398 ^c^	0.022 ^c^
SCWT-color word score	30.66 (11.55)	33.36 (10.66)	28.33 (11.12)	30.50 (12.71)	0.006 ^e^	0.391 ^b^	0.278 ^b^	0.350 ^c^
Category Fluency	34.53 (7.71)	40.68 (8.66)	32.48 (7.46)	36.79 (8.97)	<0.001 ^d,^*	0.005 ^c^	0.218 ^c^	0.024 ^b^
CTT-B	108.44 (36.78)	108.10 (38.30)	139.40 (102.79)	130.42 (64.93)	0.383 ^e^	0.890 ^c^	0.082 ^c^	0.127 ^c^
Similarities	12.00 (3.03)	11.68 (2.52)	10.73 (2.85)	10.53 (2.99)	0.296 ^e^	0.949 ^c^	0.024 ^c^	0.043 ^c^
Matrix Reasoning	12.26 (2.71)	11.66 (2.95)	10.95 (3.07)	10.71 (2.79)	0.799 ^e^	0.692 ^c^	0.023 ^c^	0.081 ^c^
**Attention**								
Attention test ^†^	7.57 (0.82)	7.63 (0.68)	7.30 (1.18)	7.24 (1.10)	0.615 ^e^	0.588 ^c^	0.278 ^c^	0.046 ^c^
Digit Span	11.81 (2.99)	12.10 (2.72)	11.10 (2.72)	12.06 (2.36)	0.056 ^e^	0.088 ^b^	0.250 ^c^	1.000 ^c^
**Processing speed**								
SCWT-word score	76.51 (16.24)	87.94 (17.52)	71.23 (19.164)	82.00 (20.06)	<0.001 ^d,^*	0.012 ^b^	0.120 ^b^	0.127 ^c^
SCWT-color score	58.68 (14.31)	64.81 (13.51)	54.467 (14.38)	63.29 (18.12)	<0.001 ^d,^*	0.011 ^b^	0.122 ^b^	0.599 ^b^
CTT-A	52.54 (21.14)	52.09 (18.10)	70.07 (40.46)	67.50 (36.04)	0.556 ^e^	0.726 ^c^	<0.001 ^c,^*	0.003 ^c^
Digit Symbol Substitution	11.93 (2.49)	12.18 (2.34)	9.88 (2.30)	10.65 (2.40)	0.029 ^e^	0.137 ^c^	<0.001 ^c,^*	<0.001 ^c,^*
Symbol Searching	12.76 (2.49)	12.09 (2.21)	10.50 (2.60)	10.09 (2.75)	0.615 ^e^	0.276 ^c^	0.009 ^c^	0.045
**Visuospatial ability**								
Pentagon copy ^†^	1/52	2/112	4/56	8/26	>0.999 ^a^	0.026 ^a^	0.369 ^a^	<0.001 ^a,^*
Block Design	11.57 (2.74)	10.65 (2.72)	9.87 (2.94)	10.03 (2.14)	0.187 ^e^	0.934 ^c^	0.003 ^c^	0.086 ^c^
**Memory**								
LM-I (r.s.)	33.13 (13.07)	36.54 (10.60)	27.97 (13.58)	33.68 (12.73)	0.001 ^e,^*	0.037 ^c^	0.060 ^c^	0.317 ^c^
LM-II (r.s.)	20.00 (10.08)	22.70 (8.76)	15.78 (10.59)	20.32 (10.26)	0.002 ^e,^*	0.041 ^c^	0.043 ^c^	0.357 ^c^
LM-recognition (r.s.)	24.04 (4.00)	24.70 (3.61)	22.08 (4.48)	23.62 (3.97)	0.039 ^e^	0.128 ^c^	0.015 ^c^	0.163 ^c^
VR-I (r.s.)	74.83 (16.38)	74.59 (13.18)	66.15 (21.31)	68.65 (17.62)	0.672 ^e^	0.587 ^c^	0.029 ^c^	0.138 ^c^
VR-II (r.s.)	50.57 (24.94)	53.19 (19.68)	43.62 (24.36)	47.41 (21.87)	0.084 ^e^	0.376 ^c^	0.131 ^c^	0.248 ^c^
VR-recognition (r.s.)	42.34 (4.17)	42.19 (3.79)	41.15 (4.20)	41.18 (4.07)	0.591 ^e^	0.991 ^c^	0.094 ^c^	0.179 ^c^
**Language** ^†^								
Naming	0/53	0/114	0/60	0/34	-	-	-	-
Repetition	2/51	2/112	0/60	1/33	0.592 ^a^	0.362 ^a^	0.218 ^a^	>0.999 ^a^
Verbal comprehension	2/51	1/113	1/59	1/33	0.237 ^a^	>0.999 ^a^	0.599 ^a^	0.408 ^a^

^a^ Fisher Exact Test; ^b^
*t*-test; ^c^ Mann–Whitney U test; ^d^ ANCOVA (adjusted for years of education as covariates); ^e^ rank analysis of covariance (adjusted for years of education as covariates); ^†^ Subtest of Mini-Mental State Examination (MMSE); * *p* < 0.002 (Bonferroni correction for multiple comparisons); Abbreviations: please see Table 1 and Table 2; LED, levodopa equivalent dose; H&Y stage, Hoehn and Yahr Staging Scale; DSS, Digit symbol substitution; r.s., raw score; LM-I, immediate memory of the logical memory; LM-II, delayed memory of the logical memory; VR-I, immediate memory of the visual reproduction; VR-II, delayed memory of the logical memory; M-WCST-C and M-WCST-P indicate the achieved categories and perseverative, respectively. ^w^ Comparisons between male NCs and female NCs; ^x^ Comparisons between male PD and female PD; ^y^ Comparisons between male NCs and male PD; ^z^ Comparisons between female NCs and female PD.

**Table 4 brainsci-11-01331-t004:** Correlations between the clinical characteristics and the neurocognitive performances in male PD.

Male PD	Onset Age		Disease Duration		LED		Stages	
	*r*	*p*	*r*	*p*	*r*	*p*	*r*	*p*
**Executive function**								
M-WCST-C	−0.143	0.275	0.003	0.980	−0.122	0.353	−0.222	0.088
M-WCST-P	0.293	0.023	−0.155	0.238	0.065	0.622	0.188	0.150
SCWT-color word score	−0.224	0.085	−0.120	0.361	−0.197	0.131	−0.116	0.377
Category Fluency	−0.042	0.753	−0.024	−0.855	−0.166	0.205	−0.151	0.248
CTT-B	0.466	<0.001 *	0.115	0.383	0.003	0.980	0.294	0.023
Similarities	0.166	0.206	−0.104	0.428	−0.196	0.133	0.073	0.577
Matrix Reasoning	0.010	0.942	−0.010	0.941	−0.024	0.856	0.096	0.466
**Attention**								
Attention test ^†^	−0.224	0.085	−0.080	0.541	0.123	0.348	−0.144	0.271
Digit Span	−0.110	0.402	−0.013	0.922	−0.208	0.112	0.070	0.596
**Processing speed**								
SCWT-word score	−0.257	0.047	−0.071	0.590	−0.156	0.235	−0.154	0.241
SCWT-color score	−0.257	0.048	−0.160	0.222	−0.145	0.267	−0.269	0.037
CTT-A	0.446	<0.001 *	0.014	0.913	0.087	0.507	0.292	0.024
Digit Symbol Substitution	−0.089	0.498	0.043	0.745	−0.235	0.071	−0.235	0.070
Symbol Searching	−0.147	0.264	0.054	0.684	−0.214	0.100	0.000	>0.999
**Visuospatial ability**								
Pentagon copy ^†^	−0.177	0.756	0.119	0.366	−0.005	0.969	0.075	0.566
Block Design	−0.206	0.114	−0.058	0.658	−0.144	0.271	0.032	0.806
**Memory**								
LM-I (r.s.)	−0.103	0.435	−0.230	0.077	−0.075	0.567	−0.144	−0.272
LM-II (r.s.)	−0.163	0.215	−0.216	0.098	−0.110	0.401	−0.160	0.221
LM-recognition (r.s.)	−0.232	0.075	−0.018	0.889	−0.061	0.644	−0.040	0.760
VR-I (r.s.)	−0.409	0.001 *	−0.0110	−0.939	−0.045	0.730	−0.100	0.446
VR-II (r.s.)	−0.347	0.007	−0.224	0.086	−0.160	0.223	−0.113	0.391
VR-recognition (r.s.)	−0.285	0.027	−0.045	0.729	−0.046	0.729	−0.174	0.183
**Language** ^†^								
Naming	-	-	-	-	-	-	-	-
Repetition	-	-	-	-	-	-	-	-
Verbal comprehension	−0.377	0.003	0.070	0.596	0.053	0.686	0.074	0.577

Abbreviations: please see Table 1, Table 2 and Table 3. * *p* < 0.002 (Bonferroni correction for multiple comparisons); ^†^ Subtest of Mini-Mental State Examination (MMSE).

**Table 5 brainsci-11-01331-t005:** Correlations between the clinical characteristics and the neurocognitive performances in female PD.

Female PD	Onset Age		Disease Duration		LED		Stages	
	*r*	*p*	*r*	*p*	*r*	*p*	*r*	*p*
**Executive function**								
M-WCST-C	−0.566	<0.001 *	−0.254	0.146	−0.236	0.178	−0.156	0.377
M-WCST-P	0.303	0.082	0.472	0.005	0.428	0.011	0.287	0.100
SCWT-color word score	−0.338	0.051	−0.276	0.114	−0.312	0.072	−0.237	0.178
Category Fluency	−0.370	0.031	−0.429	0.011	−0.335	0.053	−0.296	0.090
CTT-B	0.406	0.017	0.207	0.240	0.146	0.410	0.155	0.383
Similarities	−0.008	0.963	0.226	0.199	0.171	0.333	0.348	0.044
Matrix Reasoning	−0.053	0.766	0.383	0.026	0.276	0.114	0.099	0.576
**Attention**								
Attention test ^†^	−0.238	0.175	−0.062	0.728	−0.163	0.356	−0.029	0.870
Digit Span	0.026	0.883	−0.016	0.930	−0.017	0.923	0.335	0.053
**Processing speed**								
SCWT-word score	−0.517	0.002	−0.251	0.152	−0.226	0.152	−0.080	0.652
SCWT-color score	−0.405	0.018	−0.231	0.188	−0.231	−0.188	−0.160	0.367
CTT-A	0.427	0.012	0.094	0.596	0.094	0.596	0.121	0.497
Digit Symbol Substitution	−0.104	0.557	0.021	0.906	0.021	0.906	0.062	0.727
Symbol Searching	−0.165	0.351	0.020	0.910	0.020	0.910	0.013	0.943
**Visuospatial ability**								
Pentagon copy ^†^	0.018	0.921	−0.065	0.717	0.002	0.992	0.008	0.965
Block Design	−0.055	0.759	0.188	0.286	0.154	0.384	0.091	0.610
**Memory**								
LM-I (r.s.)	−0.358	0.037	−0.039	0.828	0.051	0.774	0.239	0.173
LM-II (r.s.)	−0.432	0.011	0.097	0.584	0.081	0.649	0.199	0.259
LM-recognition (r.s.)	−0.445	0.008	0.024	0.892	0.049	0.782	−0.127	0.473
VR-I (r.s.)	−0.331	0.056	−0.116	0.512	−0.034	0.850	−0.047	0.792
VR-II (r.s.)	−0.367	0.033	−0.224	0.203	0.023	0.899	−0.092	0.606
VR-recognition (r.s.)	−0.555	0.001 *	−0.050	0.777	−0.076	0.671	−0.025	0.887
**Language** ^†^								
Naming	-	-	-	-	-	-	-	-
Repetition	0.002	0.990	0.149	0.401	0.169	0.343	−0.123	0.487
Verbal comprehension	−0.154	0.384	0.056	0.755	−0.023	0.897	0.044	0.803

Abbreviations: please see Table 1, Table 2 and Table 3. * *p* < 0.002 (Bonferroni correction for multiple comparisons); ^†^ Subtest of Mini-Mental State Examination (MMSE).

**Table 6 brainsci-11-01331-t006:** Multiple linear regression analyses in male PD with various tests.

Male PD	Color Trails Test-Part B			Stroop Color and Word Test—Color Score			Color Trails Test-Part A			Digit Symbol Substitution		
	ΔR^2^	β Value	*p*-Value	ΔR^2^	β Value	*p*-Value	ΔR^2^	β Value	*p*-Value	ΔR^2^	β Value	*p*-Value
Model 1	0.306			0.231			0.226			0.162		
Age, y		0.554	<0.001		−0.364	0.003		0.826	<0.001		−0.076	0.535
Education, y		−0.054.	0.629		0.336	0.005		−0.192	0.100		0.400	0.002
Model 2	0.048			0.058			0.057			0.065		
Age, y		0.522	<0.001		−0.329	0.006		0.442	<0.001		−0.039	0.745
Education, y		−0.067	0.537		0.351	0.003		−0.206	0.070		0.415	0.001
Stages		0.222	0.046		−0.245	0.036		0.241	0.036		−0.258	0.035
	Total R^2^ = 0.354 *p* ≤ 0.001			Total R^2^ = 0.289 *p* ≤ 0.001			Total R^2^ = 0.283 *p* ≤ 0.001			Total R^2^ = 0.227 *p* = 0.002		

## Data Availability

The data that support the findings of this study are available from the corresponding author, upon reasonable request.

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
