# Peer review of "The Impact of Sex on the Neurocognitive Functions of Patients with Parkinson’s Disease"

_brainsci, 2021, doi:10.3390/brainsci11101331_

Round 1

Reviewer 1 Report

Authors intended to compare neurocognitive functions of patients with Parkinson's Disease depending on the sex. Authors found differences regarding neurocognitive functions in different sexes in healthy controls, the differences were not found in Parkinson's Disease (PD). The study is based on relatively small group of PD patients. The heterogeneity of the presented PD group based on basic examination may be jeopardized by such factors as patients with PSP-P, who can easily be mistaked with PD in the early stages, especially when examined using non specific tools. Additionally no examination based on the levels of hormones is performed in this study. This makes it difficult to obtain any conclusions based on this study.

Reviewer 2 Report

The paper by Chen et al focuses on the role of gender differences in the decline of cognitive function in Parkinson’s Disease (PD). From a demographic point of view, PD is known to affect males more than females (the ratio is about 2:1). There is no clear reason of why this happens. Hypothesis range from a neuroprotective role of estrogens in women and a concomitant, increased susceptibility to cardiovascular disease in men. After disease onset, there have been studies supporting the idea that gender plays a role in neurocognitive function decline. As opposed to this, data analyzed in this paper do not support such theory. The data presented in this paper seems solid, including the choice of subjects, normal controls and PD patients. What is confusing, it is the way the data is presented. The style of the article is verbose and repetitive. Therefore, what this paper would need, it would be to be rewritten and edited to make the reading easier and more effective.

In particular:

  • I would add a final sentence in the abstract summarizing the main findings.
  • I would summarize from line 50 to 72 in page 2 by simply stating that there are discrepancies between studies. Most of those concepts are repeated in the discussion
  • Line 76 seems to contradict line 68 in page 2. It is not clear if in this study only early-stage patients were used, or the paper is citing previous articles.
  • The discussion paragraph is long and repetitive. It needs to shorter and with a more effective style. In addition, the authors do not seem to offer a conclusion or to take a stand on why this study does not corroborate past investigations. Therefore, the reader is left to wonder if there are consistent differences between sex and cognitive decline in PD or not.
  • Maybe remove very old citations (before 2000).

Reviewer 3 Report

With an increasing aging population, the question of neurodegenerative diseases and the precise neurocognitive obstructed abilities is of major importance. As underlined by the authors, as the scientific literature is increasing on the cognitive deficits accompanying the development of the Parkinson disease, there are still few studies on the influence of gender on these deficits and the available results are inconsistent. For these following reasons, every study increasing knowledge in this domain is welcome. Overall, I think that the paper is well constructed, the methodology robust and the results well presented and discussed. However, I would have a general remark regarding the interpretations of the results and that could maybe help to clarify the paradoxical results underlined throughout the literature by the authors. As the authors may know, I think to the growing number of publications focusing on the socioaffective troubles appearing (sometimes before the cognitive troubles) along with the development of the disease. To me, this is now difficult to interpret neurocognitive results by keeping these interpretations purely on the cognitive domain. As there are numerous interactions between affective and cognitive processes and that the question of gender has also been considered in the affective domain in Parkinson Disease, I thought it would be useful to complete this research to discuss these results by incorporating information's from the socioaffective research.

Round 2

Reviewer 1 Report

The explanations made by the authors do not address all of my concerns e.g.:

"All participants' 109 exclusion criteria were as followsed: with atypical features of parkinsonism"

What examinations were performed to exclude atypical features of parkinsonism?

Were sufficient tools implemented to avoid including PSP-P patients to the study?

The heterogeneity of the examined groups accompanied by questionable methodology jeopardizes any conclusion.

Reviewer 3 Report

OK
